# The transmembrane protein Syndecan is required for stem cell survival and maintenance of their nuclear properties

Buffy L. Eldridge-Thomas[1], Jerome G. Bohere[1☉], Chantal Roubinet[2☉],
Alexandre Barthelemy[1¤], Tamsin J. Samuels[1,3], Felipe Karam Teixeira[1,3],
Golnar Kolahgar[1]*

**1** Department of Physiology, Development and Neuroscience, University of Cambridge, Cambridge, United Kingdom, **2** Université de Rennes, CNRS, INSERM, IGDR (Institut de Génétique et Développement de Rennes), UMR 6290, ERL U1305, Rennes, France, **3** Department of Genetics, University of Cambridge, Cambridge, United Kingdom

☉ These authors contributed equally to this work.
¤ Current address: Institut Curie, Laboratory of Genetics and Developmental Biology, PSL Research University, INSERM U934, CNRS UMR3215, Paris, France
* gk262@cam.ac.uk

## Abstract

Tissue maintenance is underpinned by resident stem cells whose activity is modulated by microenvironmental cues. Using *Drosophila* as a simple model to identify regulators of stem cell behaviour and survival *in vivo*, we have identified novel connections between the conserved transmembrane proteoglycan Syndecan, nuclear properties and stem cell function. In the *Drosophila* midgut, Syndecan depletion in intestinal stem cells results in their loss from the tissue, impairing tissue renewal. At the cellular level, Syndecan depletion alters cell and nuclear shape, and causes nuclear lamina invaginations and DNA damage. In a second tissue, the developing *Drosophila* brain, live imaging revealed that Syndecan depletion in neural stem cells results in nuclear envelope remodelling defects which arise upon cell division. Our findings reveal a new role for Syndecan in the maintenance of nuclear properties in diverse stem cell types.

## Author summary

Most tissues are built and maintained by stem cells that divide to produce correct cell types in response to environmental cues. In high turnover tissues, such as the intestine, a fundamental question is how stem cells themselves are maintained over time in the face of physical and chemical damage. Here, we show that the evolutionary conserved adhesion protein Syndecan contributes to intestinal stem cell survival by promoting the maintenance of nuclear properties in *Drosophila*. We show that this novel Syndecan-nucleus connection is not unique to intestinal stem cells and operates in neural stem cells of the developing brain too, affecting how fast these cells can divide. In the intestine however, due to higher sensitivity to Syndecan loss, stem cells develop DNA damage and fail to produce new intestinal cells, compromising cell production. Overall, our work highlights a novel stem cell survival mechanism that may be especially relevant in tissues with high mechanical perturbations.

**Data availability statement:** All relevant data are within the manuscript and its supporting information files.

**Funding:** This work was funded by a Wellcome Trust (https://wellcome.org) and Royal Society (https://royalsociety.org) Sir Henry Dale fellowship to GK (206208/Z/17/Z); a Wellcome Trust PhD studentship to BLET (102175/B/13/Z); funding from the University of Cambridge (https://www.cam.ac.uk; by the School of Biological Sciences to JGB, and by a Herchel Smith Postdoctoral Fellowship to TJS); a Wellcome Trust and Royal Society Sir Henry Dale Fellowship (206257/Z/17/Z) and a Human Frontier Science Program (https://www.hfsp.org; CDA-00032/2018) to FKT; European Union's Horizon 2020 Research and Innovation Programme under the Marie Skłodowska-Curie actions (https://www.h2020.md/en/content/marie-skłodowska-curie-actions, Grant Agreement No 899546) to CR; Agence Nationale de la Recherche (https://anr.fr, Le Borgne Lab (host to CR): ANR-20-CE13-0015) and Cancer Research UK Discovery Programme (https://www.cancerresearchuk.org, Baum Lab (host to CR): Grant Award 28276). The funders had no role in study design, data collection and analysis, decision to publish, or preparation of the manuscript.

**Competing interests:** The authors have declared that no competing interests exist.

## Introduction

Most tissues are renewed by resident stem cells, capable of producing new specialised cells when required. Mechanisms promoting stem cell maintenance include genome protection, induced quiescence and stem cell self-renewal through asymmetric cell divisions [1]. Deregulation of these mechanisms can lead to acquisition of mutations and cancer development or stem cell loss and tissue attrition [1].

Stem cell survival and fate decisions are influenced by the basement membrane, a specialised extracellular matrix whose complex structure and composition varies dynamically with physiological context [2–4]. Whilst *in vitro* systems are often used to study stem cell behaviour, they incompletely recapitulate this extracellular environment, highlighting the ongoing value of *in vivo* models to further our understanding of stem cell biology and disease. To investigate stem cells in their native milieu, we turned to the simple, genetically tractable *Drosophila*. The adult *Drosophila* gut is maintained throughout life by resident stem cells [5,6] (Fig 1A-A'), allowing long term assessment of stem cell maintenance and tissue-level effects, while the developing larval brain contains rapidly dividing neural stem cells highly amenable to live imaging, facilitating a more dynamic understanding of cell biological events. While testing the role of basement membrane receptors in stem cell activity, we investigated the single *Drosophila* ortholog of Syndecan (Sdc) [7].

Sdc proteins are transmembrane proteoglycans bearing heparan and chondroitin sulfate chains on their extracellular domain, via which they bind to basement membrane components including Laminin and Collagen IV [8,9]. Mammals possess four *sdc* genes with varying extracellular domains and a highly conserved small cytoplasmic domain involved in protein-protein interactions, including direct and indirect association with the microtubule and actin cytoskeletons [9,10]. Sdc proteins play important and diverse roles during development, inflammation, disease and tissue repair, through adhesion and signalling [11–14]. However, so far, little is known about Sdc's role in stem cell maintenance and tissue homeostasis.

Here we report that *Drosophila* Sdc contributes to intestinal stem cell maintenance, through pro-survival mechanisms associated with nuclear properties and genome protection. Furthermore, we identify an additional role for Sdc in nuclear envelope remodelling during asymmetric division of neural stem cells. Our findings uncover a novel connection between Sdc and nuclear properties in multiple stem cell types.

## Results

### Syndecan is required for long-term *Drosophila* intestinal stem cell maintenance

Sdc, a previously uncharacterised player in the adult *Drosophila* intestine, is expressed throughout the intestinal epithelium, in all cell types (Figs 1B, 1C and S1). To test whether Sdc contributes to intestinal cell production, we knocked down Sdc by RNAi expression in progenitor cells (Fig 1A) using the *esg^{ts} F/O* ("escargot flip out") system [15] (Methods). This system also drives heritable GFP expression in progenitor cells and their progeny, allowing assessment of progenitor maintenance and new cell production. Expression of two out of three independent RNAi lines impaired new cell production as evidenced by limited appearance of newly differentiated large GFP$^{+ve}$ cells in the tissue (Figs 1D-F, S2A and S2B) and resulted in a progressive loss of small GFP$^{+ve}$ progenitor cells (Figs 1D and S2A–S2C).

To identify in which cell type(s) of the posterior midgut Sdc is required, we performed RNAi knockdown with cell type-specific GAL4 drivers (Fig 1A and Methods). We used Sdc RNAi 3 (henceforth referred to as 'Sdc RNAi' for simplicity) as this line yielded highly effective knockdown (S2A' Fig) and has also been validated and used in other work [16,17].

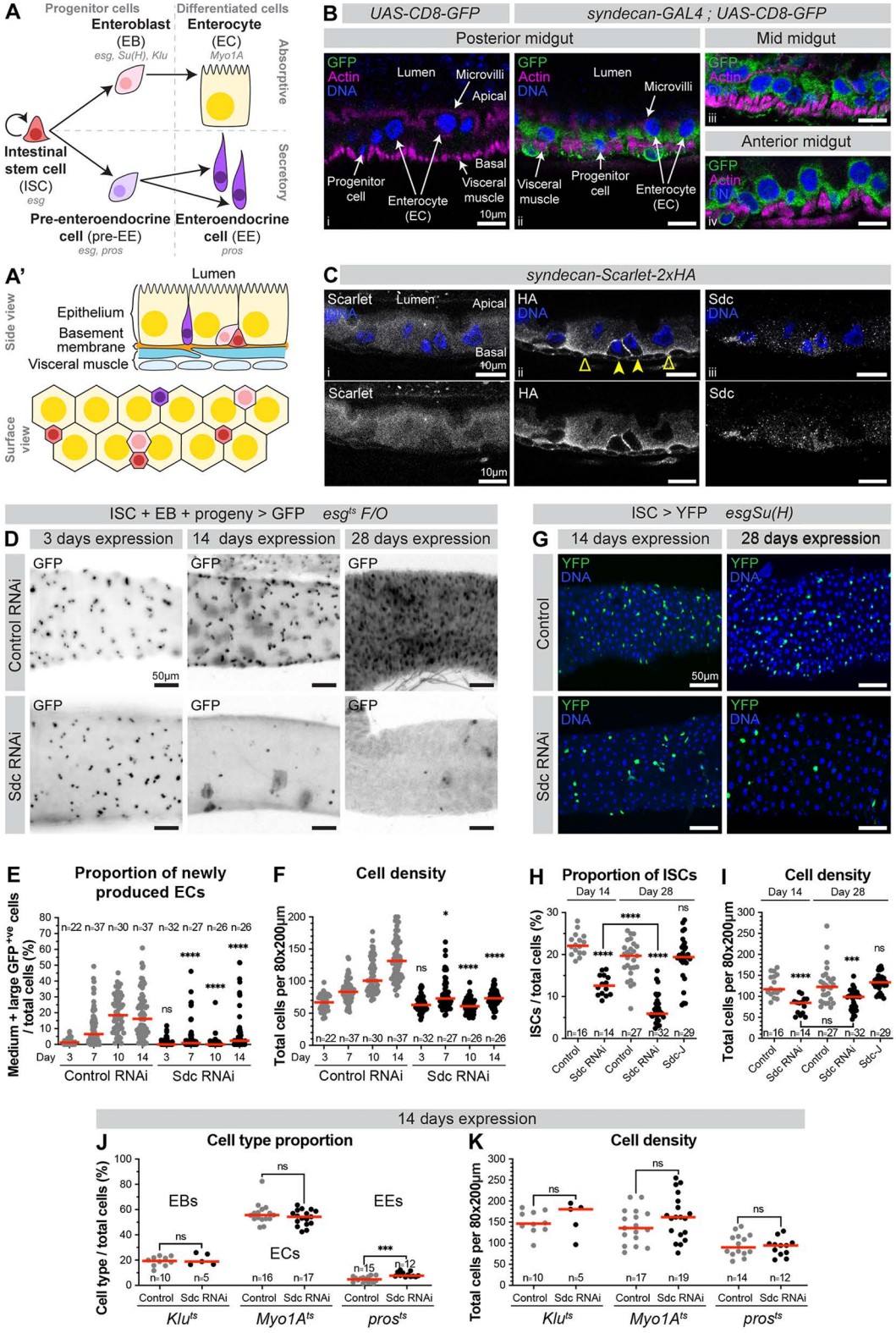

**Fig 1. Sdc is required for *Drosophila* intestinal stem cell maintenance.** (A) Intestinal cell lineages. Intestinal stem cells (ISCs) self-renew and give rise to enteroblasts (EBs) which differentiate into absorptive enterocytes (ECs); and pre-enteroendocrine cells (pre-EEs) which undergo one division before differentiating into a pair of secretory enteroendocrine cells (EEs). Progenitor cells comprise ISCs, EBs and pre-EEs. Cell type-specific genes shown in italics. (A') Side and surface view schematics of the

posterior midgut. (B) Side views of midguts carrying either *UAS-CD8-GFP* alone (i) or with *syndecan-GAL4* (ii-iv). Progenitor cells are identified by their small, basally located nuclei. ECs are identified by their large nuclei. DNA stain (blue) marks nuclei; GFP (green) reports *sdc* expression; Phalloidin (magenta) marks the visceral muscle and EC microvilli. (C) Side views of midguts expressing endogenous Sdc tagged with Scarlet and HA at its C-terminus. ECs (empty arrowheads) show diffuse, basal Sdc signal. Progenitors/EEs (filled arrowheads) show Sdc enrichment. DNA stain (blue) marks nuclei; Scarlet (i), anti-HA (ii) and anti-Sdc (iii) (all white) mark Sdc protein. (D) Surface views of midguts expressing control RNAi or Sdc RNAi using the *esg^{ts} F/O* system. GFP (black) marks small progenitor cells and their progeny. (E) Proportion of medium and large GFP^{+ve} cells (corresponding to newly produced differentiating and terminally differentiated ECs, respectively), and (F) Cell density. n = number of guts, from three replicates. (G) Surface views of midguts expressing Sdc RNAi in ISCs using the *esgSu(H)* system. DNA stain (blue) marks the nuclei of all cells; YFP (green) marks ISCs. (H) ISC proportion and (I) Cell density. n = number of guts, from two (Day 14) or three (Day 28) replicates. (J) Cell type proportion and (K) Cell density of midguts expressing Sdc RNAi in EBs (*Klu^{ts}*), ECs (*Myo1A^{ts}*) or pre-EEs and EEs (*pros^{ts}*). n = number of guts, from one (EBs) or three (ECs & EEs) replicates. ns: not significant, *: $p < 0.05$, ***: $p < 0.001$, ****: $p < 0.0001$.

We employed *w^{1118}* as the control (referred to as 'Control') to match the genetic background in which Sdc RNAi 3 was originally constructed [18], as this is known to impact intestinal stem cell dynamics during aging [19]. We monitored the proportion of each cell type to assess whether Sdc is required for its maintenance. In addition, since failure in new cell production is accompanied by low epithelial cell density (Fig 1F), we measured cell density as a proxy for new cell production. Sdc knockdown in ISCs resulted in a progressive reduction in ISC proportion (Fig 1G and 1H). This ISC loss was associated with reduced cell density (Fig 1I), presumably due to a failure in new cell production (Fig 1E). ISC loss was also observed using two further Sdc RNAi lines (S2D Fig). Fly survival was not compromised (S3 Fig), consistent with other reports using alternative modes of ISC elimination [20,21].

Knockdown in other cell types did not perturb their proportion (Fig 1J), except for EEs which became more abundant. Based on known roles of Sdc in other *Drosophila* tissues [22–25], this could be due to Sdc modulating Slit/Robo signalling involved in EE specification [26,27]. Nevertheless, across all knockdowns in cell types other than ISCs, cell density was comparable to control guts (Fig 1K), suggesting Sdc is dispensable in other epithelial cell types. We then tested whether Sdc overexpression in ISCs would induce an opposite effect on ISC proportion or cell density, but it did not (Fig 1H and 1I). Altogether, we have identified Sdc as a previously unrecognised but important factor required for stem cell maintenance in the adult *Drosophila* intestine.

## Sdc-depleted intestinal stem cells are partially lost through apoptosis and display aberrant cell shapes

ISCs could be eliminated from the tissue by differentiation or cell death. Differentiation is ruled out by the reduction in differentiated cells upon expression of Sdc RNAi with the *esg^{ts} F/O* lineage tracing system (Figs 1D, 1E and S2; compare to control guts where production of large GFP^{+ve} ECs is clear). To test whether ISC loss occurs by apoptosis, a common form of programmed cell death, we expressed the inhibitor of apoptosis, DIAP1 [28], in these cells. DIAP1 expression partially rescued (~50%) ISC loss caused by Sdc depletion and did not affect ISC numbers in control guts (Fig 2A and 2B). Thus, some ISCs depleted of Sdc are eliminated by apoptosis.

We then sought to determine the cellular changes induced by loss of Sdc, which could cause ISC elimination. Observation of labelled Sdc-depleted ISCs at high magnification revealed striking changes in cell shape. *Drosophila* ISCs are normally small and triangular shaped [6,29,30] (Fig 2Ci-ii). In contrast, Sdc-depleted ISCs showed a variety of morphological defects including larger size, cytoplasmic protrusions, and occasional blebs (Fig 2Ciii-vii). Quantification of ISC area and shape revealed that, compared to controls, ISCs expressing

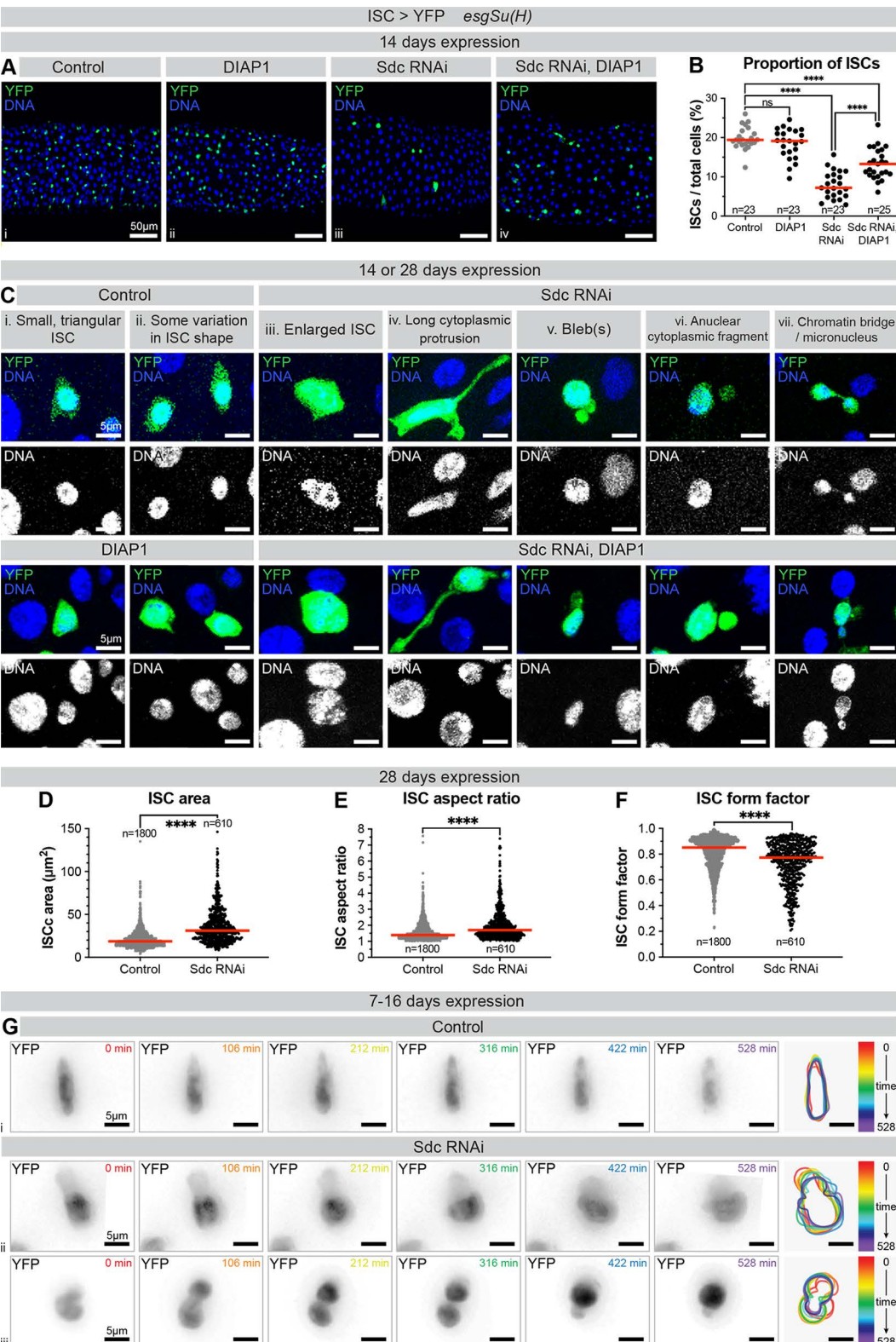

**Fig 2. Sdc-depleted ISCs adopt abnormal cell shapes.** (A) Surface views of midguts expressing +/− DIAP1 and +/− Sdc RNAi in ISCs. DNA stain (blue) marks the nuclei of all cells; YFP (green) marks ISCs. (B) ISC proportion. n = number of guts, from three replicates. (C) Surface views of ISCs. DNA stain (blue/white) marks nuclei; YFP (green) marks ISCs. Images are from either 14 or 28 days expression with *esgSu(H)*. (D) ISC cytoplasmic area; (E) ISC aspect ratio and (F) ISC form

factor. n = number of ISCs analysed, from ≥27 guts, from three replicates. (G) Time lapse of control or Sdc RNAi-expressing ISCs. YFP (black) marks ISCs. Right panels show colour-coded snapshots of the cell outline during the time lapse. ns: not significant, ****: p < 0.0001.

Sdc RNAi were larger (Fig 2D), more elongated (Fig 2E) and more convoluted (Fig 2F). ISC-specific DIAP1 expression, with or without concomitant Sdc depletion, also resulted in larger cells (S4A Fig). In all genotypes, larger cells were more convoluted (S4B-E Fig), with a stronger effect seen when Sdc RNAi was also expressed (S4C and S4E Fig). Focussing on these larger (>50 μm$^2$) cells, we observed similar deformations when Sdc RNAi was expressed both with and without DIAP1 (S4F and S4G Fig), suggesting that these cell shape changes are not primarily caused by apoptosis.

Abnormal cell shape and size might reflect changes in the cortical cytoskeleton and cellular dynamics. We therefore turned to live imaging of fluorescently labelled ISCs in *ex vivo* guts to compare dynamic cellular behaviour over approximately eight hours (>400 ISCs imaged across >14 guts for each genotype). Whilst most control ISCs remained static (Fig 2Gi), Sdc RNAi-expressing ISCs displayed more dynamic protrusions and shape changes (Fig 2Gii). In addition, rare Sdc RNAi-expressing ISCs appeared to attempt but fail in cell division (Fig 2Giii), a behaviour that we have never observed in controls. We concluded that Sdc modulates stem cell shape and that its loss can contribute to changes in cellular morphology compatible with defective divisions.

## Sdc is dispensable for intestinal stem cell abscission but prevents nuclear lamina invaginations and DNA damage

How could Sdc interfere with stem cell morphology and maintenance? Sdc-4 links the abscission machinery to the plasma membrane in mammalian cultured cells, with Sdc-4 depletion delaying abscission or more rarely causing abscission failure, generating binucleate cells [31]. To explore whether Sdc depletion from proliferating *Drosophila* ISCs impairs their resolution into two separate daughter cells and causes them to become binucleate prior to their elimination, we immunostained guts with Lamin B to label the nuclear compartment and α-catenin to label the cell perimeter (Fig 3A). To confirm that our approach could detect binucleate cells, we used RNAi to knockdown the cytokinesis-promoting Aurora B [32] (Fig 3A). We did not detect any binucleate stem cells upon Sdc knockdown (n>100 ISCs examined across >20 guts) (Fig 3A), indicating that Sdc is not required for ISC abscission.

We noted, however, that unlike control ISC nuclei which appeared approximately spherical, with a smooth, round nuclear lamina (Fig 3A), the nuclei of Sdc-depleted ISCs presented frequent nuclear lamina invaginations (Fig 3A and 3C) and aberrant nuclear shapes, with more elongated and lobed nuclei (Fig 3D and 3E). These phenotypes were also seen with other Sdc RNAi lines and when DIAP1 was co-expressed with Sdc RNAi (S5A Fig), indicating that they were neither an off-target effect of the RNAi nor a secondary effect of apoptotic cell death.

We reasoned that Sdc depletion could impair a relay from the plasma membrane to the nucleus, which would in turn alter nuclear properties. The Linker of Nucleoskeleton and Cytoskeleton (LINC) complex is a conserved multiprotein complex that connects the cytoskeleton with the nucleoskeleton, allowing transmission of mechanical force from the cell surface to the nucleus, regulating nuclear deformation, positioning and functions [33,34]. The *Drosophila* LINC complex consists of two KASH proteins, Klarsicht (Klar) and Msp300, and two SUN proteins, Klaroid (Koi) and the testes-specific SPAG4 (Fig 3B) [33]. We hypothesised that Sdc might act via the LINC complex and thus tested whether individual knockdowns

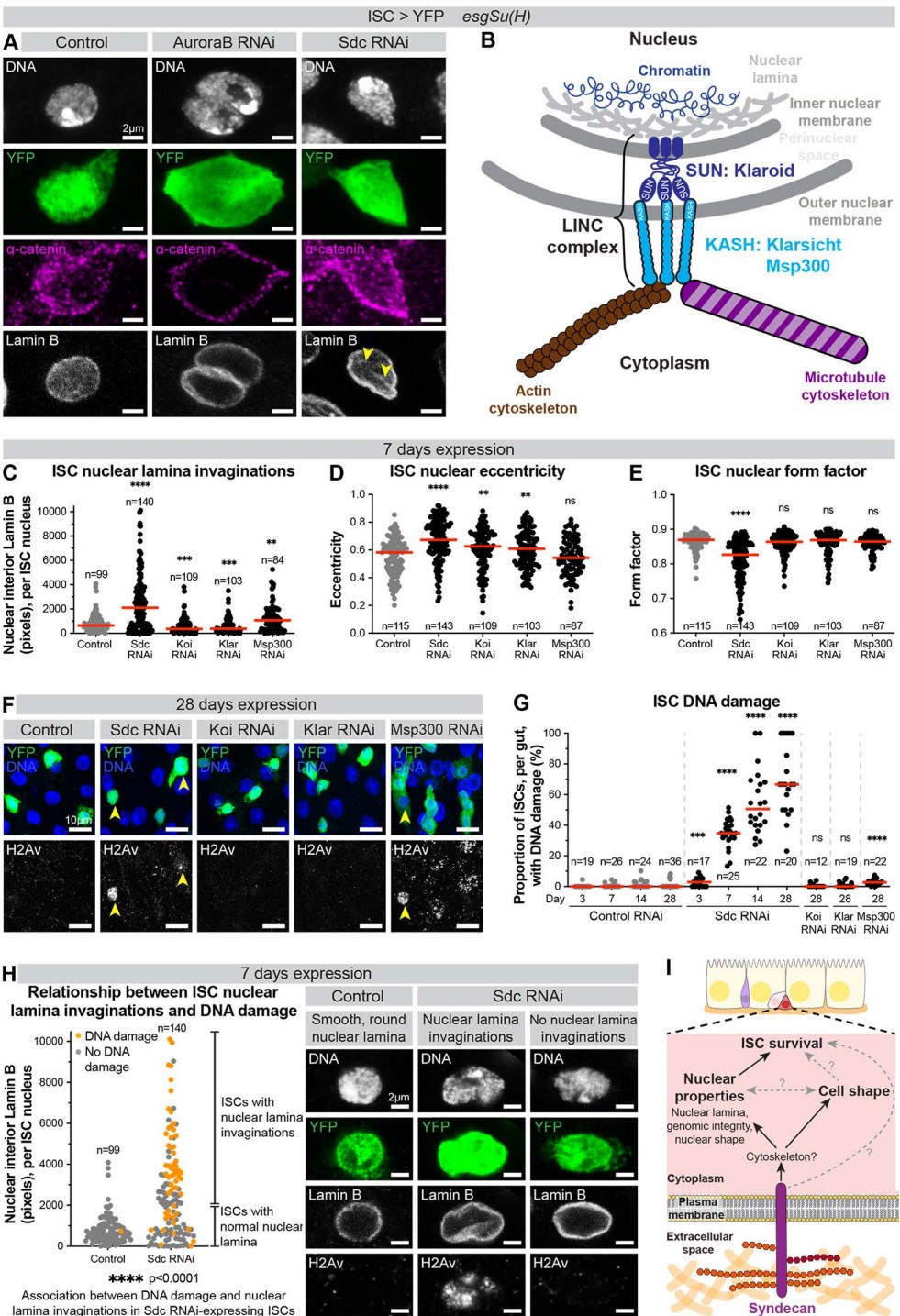

**Fig 3. Sdc prevents nuclear lamina invaginations and DNA damage.** (A) Control ISC, and ISC expressing Aurora B RNAi or Sdc RNAi. DNA stain (white) marks nuclei; YFP (green) marks ISCs; anti-α-catenin (magenta) marks cell junctions; anti-Lamin B (white) marks nuclear lamina. Yellow arrowheads indicate nuclear lamina invaginations. (B) The *Drosophila* LINC complex. (C) ISC nuclear lamina invaginations (Lamin B pixels located in the nuclear interior), (D) ISC nuclear eccentricity, and (E) ISC nuclear form factor. n = number ISCs analysed, from a minimum of 17 guts per genotype, from a minimum of two replicates. (F) Surface view images of midguts. DNA stain (blue) marks nuclei; YFP (green) marks ISCs; anti-γH2Av (white) marks DNA damage. Yellow arrowheads indicate ISCs with DNA damage. (G) Proportion of ISCs, per gut, with DNA damage. n = number of guts, from a minimum of two replicates. >100 ISCs analysed per genotype per timepoint. (H) Relationship between ISC nuclear lamina invaginations and

DNA damage. Each dot represents an individual ISC, coloured orange if ISC has DNA damage and grey if ISC does not. ISCs are defined as having nuclear lamina invaginations if there are ≥2000 Lamin B pixels in the nuclear interior. n = number ISCs analysed, from ≥22 guts per genotype, from four replicates. Example ISCs, with and without nuclear lamina invaginations and DNA damage. DNA stain (white) marks nuclei; YFP (green) marks ISCs; anti-Lamin B (white) marks the nuclear lamina; anti-γH2Av (white) marks DNA damage. (I) Model representing the role of Sdc in *Drosophila* ISCs. ns: not significant, **: p < 0.01, ***: p < 0.001, ****: p < 0.0001.

of Klar, Koi and Msp300 in ISCs recapitulated the lamina invaginations and nuclear shape changes seen upon Sdc depletion. The knockdown efficacy of all RNAi lines used here has previously been validated by qPCR and/or immunostaining [35–38]. Only Msp300 knockdown resulted in increased lamina invaginations compared to control nuclei, but not to the same extent as Sdc knockdown (Figs 3C and S5A). Knockdown of Klar or Koi produced only modest nuclear elongation (Fig 3D) and nuclear lobing was unaffected (Fig 3E).

Disruptions to the nuclear lamina are often associated with DNA damage [39,40]. Immunolabelling of DNA double strand breaks with γH2Av [41] revealed striking DNA damage acquisition in ISCs upon Sdc knockdown but not in control ISCs (Fig 3F and 3G). Increased DNA damage was also seen with a second Sdc RNAi line, albeit at a lower frequency (S5B Fig). These results suggest that Sdc affects a mechanism protecting the genome. Knockdown of individual or pairs of LINC complex components from ISCs (to exclude potential redundancy [33,34]) did not recapitulate the level of DNA damage seen upon Sdc depletion (Figs 3F, 3G and S5C), suggesting that Sdc function is unlikely to be fully accounted for by LINC complex proteins. It should be noted that γH2Av staining in the Msp300 RNAi condition is predominantly found in ECs, not ISCs (Fig 3F). How Msp300 loss in ISCs causes DNA damage in neighbouring cells remains to be determined, however, it is clear that this does not recapitulate the high levels of ISC-specific DNA damage seen upon Sdc RNAi knockdown (Fig 3F and 3G). Notably, DNA damage was not a secondary consequence of apoptosis (S5D Fig), and thus may instead contribute to ISC elimination.

We carefully examined the frequency of both nuclear lamina invaginations and DNA damage in a large number of ISCs (Fig 3H). DNA damage was found more frequently in Sdc-depleted ISCs with lamina invaginations compared to those without (Fig 3H), demonstrating a relationship between acquisition of DNA damage and disruption of the nuclear lamina.

Altogether, our results suggest that Sdc modulates nuclear shape and the nuclear lamina, in a manner that is not dependent on single members of the LINC complex. In the absence of Sdc, nuclear aberrations arise early, and the acquisition of DNA damage could conceivably trigger ISC elimination (Fig 3I).

## Sdc controls nuclear envelope remodelling of neural stem cells but is dispensable from female germline stem cells

Next, to test whether Sdc plays similar roles in other stem cell types, we examined female germline stem cells (fGSCs) and larval neural stem cells (neuroblasts).

In the female *Drosophila* germline (S6A Fig), Sdc depletion from fGSCs did not cause germline loss (S6A and S6B Fig), suggesting that Sdc is dispensable for fGSC survival. Moreover, we did not detect any evidence of germline stem cysts, which arise upon defective fGSC abscission [42] (S6A and S6B Fig), indicating that Sdc is not required for abscission in these cells. In addition, we did not detect any γH2Av staining in fGSCs (S6C Fig), indicating that Sdc is not required for genome protection in these cells. Overall, presumably due to low levels of expression compared to the surrounding somatic cells (S6D Fig), our results suggest Sdc is dispensable in fGSCs, supporting previous data [43].

In the larval *Drosophila* brain, which is highly amenable to live imaging [44,45], neuroblasts undergo rapid asymmetric cell division, generating a large self-renewing neuroblast with a large nucleus and a smaller differentiating ganglion mother cell (GMC) with a small nucleus [46] (Fig 4A). Wild type neuroblasts undergo semi-closed mitosis, whereby the nuclear envelope persists throughout mitosis, dependent on the maintenance of a supporting nuclear lamina [47]. Some reservoirs of nuclear membrane are also present in the cytoplasm of the renewing neuroblast and contribute to differential nuclear growth of the daughter cells [47] (Fig 4A and 4B).

After confirming the presence of Sdc in neuroblasts (S7 Fig), we examined whether its depletion affected cell division and nuclear properties. Sdc-depleted neuroblasts exhibited prolonged cell division (Fig 4B, 4D and 4E) and had a variety of cell division, nuclear envelope and nuclear shape defects. Approximately 50% of mitotic neuroblasts expressing Sdc RNAi displayed abnormal mitotic nuclear envelope remodelling (Fig 4C-C') including ruptures (Fig 4C and 4D) and dispersion (Fig 4C and 4E) of the nuclear envelope, which appeared as early as metaphase. Outside of mitosis, Sdc-depleted neuroblasts had aberrant nuclear shape and size (Fig 4F and 4G) and abnormal cell size (Fig 4H).

Furthermore, Sdc depletion resulted in abnormal ratio in nuclear size between the renewed neuroblast and the newly formed GMC (Fig 4I). This was sometimes caused by mispositioning of the cleavage furrow and abnormal nuclear membrane partitioning between the daughter cells, resulting in abnormally large GMC nuclei (Fig 4D), and in other cases was caused by impaired nuclear growth, resulting in abnormally small neuroblast nuclei (Fig 4E). These defects in nuclear size ratio were predominantly associated with divisions where the nuclear envelope was either ruptured or completely dispersed (Fig 4I), suggesting that mitotic nuclear envelope remodelling defects induced by Sdc knockdown might have a causative effect on nuclear size ratio. These defects did not elicit detectable levels of DNA damage (S8A-C Fig). Furthermore, brain lobe sizes from larvae with Sdc-depleted neuroblasts were comparable to control brains (S8D Fig), suggesting that Sdc depletion does not affect neuroblast ability to generate differentiated progeny.

Thus, in at least two different somatic stem cell types, in the adult gut and developing brain, the transmembrane protein Sdc modulates stem cell nuclear properties, with tissue-specific impacts on cell division and tissue renewal.

## Discussion

Stem cell activity underpins tissue renewal and disease, and stem cells are highly regulated by the complex, dynamic environment in which they reside. The adult *Drosophila* gut has proven an excellent *in vivo* system allowing functional characterisation of proteins involved in basement membrane adhesion and mechanotransduction, from the molecular to tissue level [48–58]. Here we have identified previously unknown connections between the conserved transmembrane proteoglycan Sdc, nuclear properties and stem cell behaviour (S9 Fig).

Our temporal analysis of *Drosophila* ISCs depleted of Sdc revealed early nuclear lamina defects (Fig 3A and 3C) and gradual acquisition of DNA damage (Fig 3G) followed by ISC loss (Fig 1G and 1H) suggesting that Sdc promotes nuclear lamina organisation and safeguards genomic integrity to protect stem cells. How could Sdc control nuclear properties and cell survival? Sdc could prevent initiation of cell death programmes which subsequently alter the nucleus [59]. However, when we blocked apoptosis nuclear abnormalities were still observed (S5A and S5D Fig), arguing against such a mechanism.

Alternatively, Sdc could preserve nuclear properties, which in turn promote stem cell survival. In the female *Drosophila* germline, mutations which deform the nuclear lamina

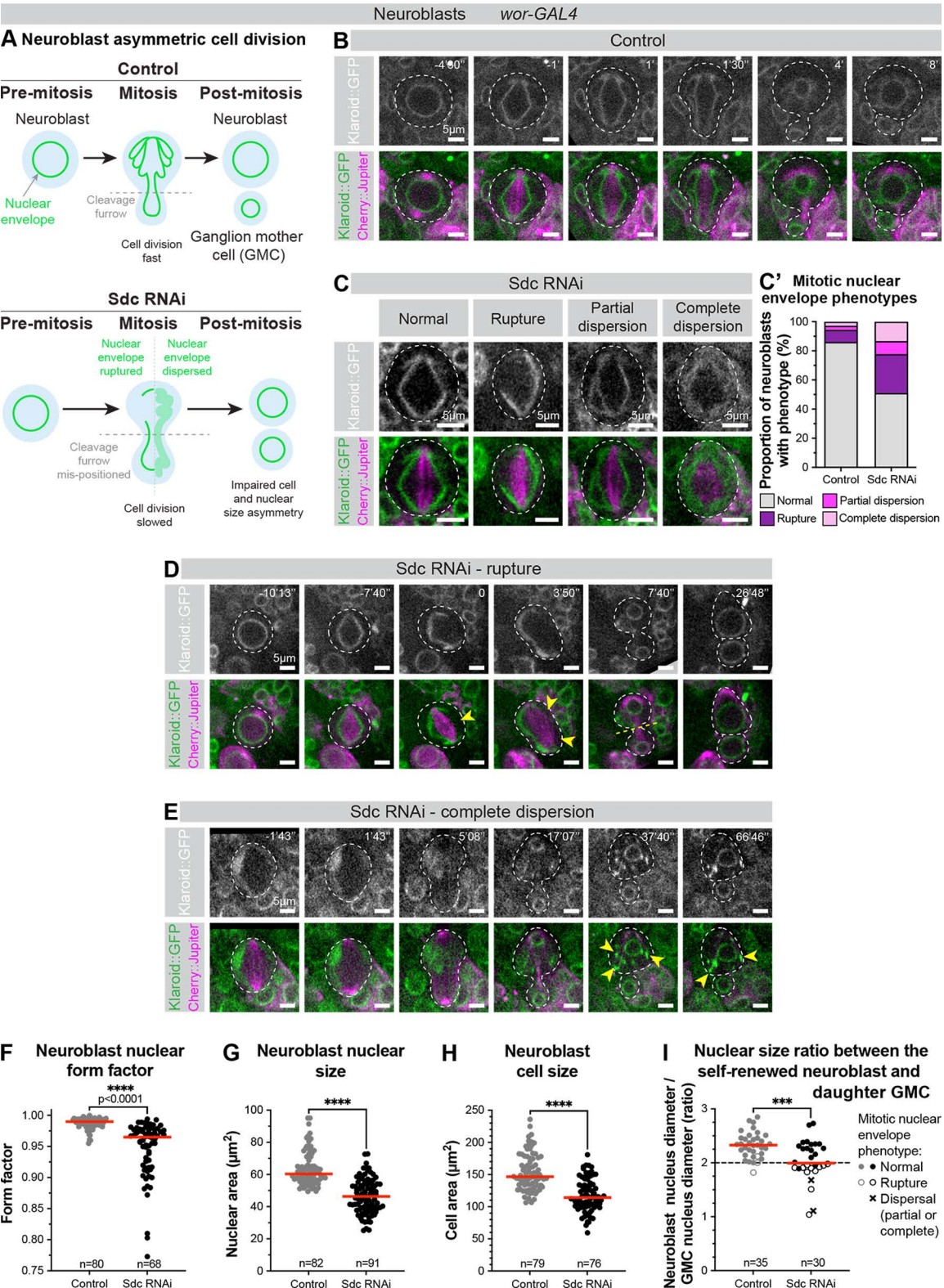

**Fig 4. Sdc knockdown in neural stem cells alters nuclear envelope remodelling and asymmetric cell division.** (A) *Drosophila* neuroblast division. (B-C & D-E) Klaroid::GFP (white/green) marks inner nuclear membrane; Cherry::Jupiter (magenta) marks mitotic spindle. Neuroblast outlined with white dashed line. Time stamp in minutes and seconds, with anaphase onset defined as time 0. (B) Time lapse of a control mitotic neuroblast. Representative time lapse from 36 neuroblasts from 4 independent experiments (20

brains total). (C) Example images of mitotic neuroblasts expressing Sdc RNAi with mitotic nuclear envelope phenotypes. Representative images from 45 neuroblasts from two independent experiments (7 brains total). (C') Prevalence of mitotic nuclear envelope phenotypes. n numbers same as B&C. (D) Time lapse of a mitotic neuroblast expressing Sdc RNAi, with a large rupture of the nuclear envelope (yellow arrowheads). Yellow dashed line marks cleavage furrow. (E) Time lapse of a mitotic neuroblast expressing Sdc RNAi, with complete dispersion of the nuclear envelope. Arrowheads mark abnormal nuclear membrane aggregates in the cytoplasm of the renewing neuroblast. (F) Neuroblast nuclear form factor, (G) Neuroblast nuclear size and (H) Neuroblast cell size, measured during interphase. n = number of neuroblasts, from 13 brains from 3 independent experiments (control) or 7 brains from 2 independent experiments (Sdc RNAi). (I) Nuclear size ratio between the renewed neuroblast and newly formed GMC, measured at telophase. Neuroblast nuclear envelope phenotype indicated for each data point. n = number of neuroblast/GMC pairs measured, from 13 brains from 3 independent experiments (control) or 7 brains from 2 independent experiments (Sdc RNAi). ***: p < 0.001, ****: p < 0.0001.

can promote DNA damage and result in stem cell loss [60,61]. This is compatible with our results which show a strong association between nuclear lamina defects and DNA damage in Sdc-depleted ISCs (Fig 3H). However, Sdc appeared dispensable in fGSCs and was not required for their genomic integrity (S6 Fig), suggesting Sdc may have tissue-specific roles. Besides low Sdc expression levels (S6D Fig), fGSCs have a unique nuclear architecture [61] and we speculate that this might result in differences in how fGSCs control the physical integrity of their nuclei. Furthermore, while fGSC maintenance relies on cell-cell contact with somatic cells of the niche [62], there is no evidence that fGSCs are in contact with a pronounced extracellular matrix/basement membrane domain. This could also explain the different requirement for Sdc, a basement membrane binding protein, between fGCSs and ISCs. In ISCs, the proximal cause of death upon Sdc knockdown remains to be determined, although the DNA damage acquired by these cells seems a likely driving force, with physiological DNA damage and genotoxic stress known to contribute to ISC elimination [63,64]. Future work should explore the mechanistic links between Sdc depletion, DNA damage and ISC loss.

Might Sdc be required at a particular phase of the cell cycle? Sdc depletion in neuroblasts perturbs mitotic nuclear envelope remodelling and impairs cell and nuclear division, with effects seen as early as metaphase (Fig 4A and 4C-E), revealing additional roles of Sdc in cell division, divergent from those found in mammalian cell culture [31,65]. It is notable that in the gut, which is predominantly composed of post-mitotic cells, Sdc depletion was only deleterious in the cell population capable of dividing (Fig 1). Future live imaging of the *Drosophila* gut would allow dynamic assessment of ISC behaviour and analysis of whether nuclear aberrations are precipitated by cell division, as appears to be the case in neural stem cells.

As our results pointed towards a connection between Sdc and nuclear properties, we reasoned that Sdc might be involved in force transmission between the plasma membrane and nucleus [66,67]. Indeed, mammalian Sdc-1 and -4 can act as mechanosensors, transmitting force via their interaction with the cytoskeleton [68,69], although Sdc's involvement in mechanical signalling remains understudied in comparison to its involvement in chemical signalling. Excitingly, we found that knockdown of components of the LINC complex, stereotypically considered the major machinery relaying force to the nucleus, did not produce similar nuclear aberrations seen upon Sdc depletion, suggesting that Sdc might act via a LINC-independent route, or that LINC complex components may be influenced by environmental conditions. For example, Klar facilitates nuclear movement as ISCs migrate locally during tissue repair but is dispensable under unchallenged conditions when ISCs are relatively immotile [70,71] (Fig 2G).

If changes to the nucleus are not mediated via the LINC complex, we speculate that Sdc tunes the cytoskeleton, with subsequent effects on nuclear properties (Fig 3I). There is a large body of evidence showing that the cytoskeleton can influence nuclear properties [67,72–74],

and altered cytoskeletal dynamics could also drive cell shape changes induced upon Sdc knock-down (Fig 2C and 2G). Recent work in mammalian cells has uncovered a laminin-keratin link which shields the nucleus from actomyosin-mediated mechanical deformation, with keratin intermediate filaments forming a protective meshwork around the nucleus [75]. While *Drosophila* cells lack these cytoplasmic intermediate filaments, it is possible that microtubule and/or actin networks instead modulate the nucleus in a force-dependent manner, as reported in other contexts [76,77]. Notably, Sdc can directly bind tubulin [9,78] and actin-binding proteins, such as FERM domain-containing proteins [9,79]. Future work should seek to identify the precise molecular mechanism(s) via which Sdc elicits its effects at the nucleus, and whether these depend on its interaction with the extracellular matrix. Development of knockout tools and re-expression of truncated forms of the Sdc protein would help achieve this.

In conclusion, using a simple, tractable *in vivo* model, we have discovered that Sdc is required for maintenance of nuclear properties and cell function of intestinal and neural stem cells. Whilst there will be microenvironmental differences between cell types and organisms, given Sdc's high conservation, the broad relationship between Sdc and nuclear properties is likely to be upheld across organisms. This may be therapeutically relevant, especially as nuclear changes and genomic instability are hallmarks of cancer cells, and that Sdc proteins are often deregulated in cancer and inflammatory diseases [80,81].

## Materials and methods

### Fly strains

All *Drosophila* stocks were maintained at 18°C or amplified at 25°C on standard medium (cornmeal, yeast, glucose, agar, water, Nipagin food medium). Fly strains are referenced in Table 1.

We used FlyBase (release FB2023_06) to find information on phenotypes/function/stocks/gene expression etc. [82].

The fly line expressing Sdc-Scarlet-2xHA was produced by Genetivision. The following guide RNAs (gRNA) were selected for CRISPR/Cas9 mediated double strand breaks, for targeting immediately after the final codon of the *syndecan* gene: TATCTCAGGCG-TAGAACTCG CGG and GTGTGCGTATGACTGGACGA AGG. The donor template for homology directed repair was designed to contain a four-serine linker, Scarlet and a HA-GG-HA tag.

### Experimental conditions for gut analyses

All analyses were performed on mated females.

For experiments involving GAL80$^{ts}$, flies were reared at 18°C throughout embryonic, larval and pupal development, with crosses flipped every 3–4 days. Flies were collected every 3–4 days after hatching and transferred to 29°C, (which was defined as day 1) and transferred to new food every two days.

For the survival experiment, crosses of 15 *esgSu(H)* virgins and 6 $w^{1118}$ or Sdc RNAi males were set up at 18°C and flipped every 3–4 days. Upon hatching, adult flies were collected and transferred to 29°C for two days to allow mating. On day 2, flies were anesthetised for minimal time to retrieve female flies of the correct genotypes, which were transferred into vials of standard medium at a density of 12 females per vial. All vials were placed horizontally at 29°C, with vial position randomised to control for any variation in temperature or humidity in the incubator. Survival was recorded daily, and flies were transferred to fresh food every 2 days until the end of the experiment, when the final fly died.

*esg$^{ts}$ F/O* ("escargot flip out") experiments [15] were performed as in [58]. At the non-permissive temperature (18°C), GAL80$^{ts}$ which is expressed under the ubiquitous

**Table 1. Fly stocks used in this work.**

| Full genotype | Informal name | Reference/source |
|---|---|---|
| w; esg-GAL4, tub-GAL80ts, UAS-GFP/ CyO; UAS-flp, act>CD2> GAL4/ TM6B; +/ + | *esg^ts F/O* | [15] |
| +/ +; UAS-CD8-GFP; +/ +; +/ + | UAS-CD8-GFP | |
| y[1] w[*]; Mi{Trojan-GAL4.0} Sdc[MI03925-TG4.0]/ SM6a; +/ +; +/ + | *syndecan-GAL4* | BDSC 76652 |
| w; esg-GAL4 UAS-YFP/ CyO; Su(H) GBE-GAL80, tub-GAL80ts/ TM3, Sb; +/ + | *esgSu(H)* (expresses in ISCs) | Cedric Polesello |
| w; UAS-CD8-GFP, tub-GAL80ts; Klu-GAL4, UAS-H2B-RFP/ SM6a, TM6B; +/ + | *Klu^ts* (expresses in EBs) | [83] |
| +/ +; UAS-YFP/ CDY; voila-GAL4/ TM6B; +/ + | *pros^ts* (expresses in pre-EEs & EEs) | Irene Miguel-Aliaga |
| +/ +; Syndecan-Scarlet-2xHA/ (CyO); +/ +; +/ + | Sdc-Scarlet | this study |
| yw; MyoIA-GAL4, tub-GAL80ts/ Cyo; UAS-GFP/ TM6b; +/ + | *MyoIA^ts* (expresses in ECs) | Kolahgar lab |
| y[1] sc[*] v[1] sev[21]; P{y[+t7.7]; +/ +; v[+t1.8]=VALIUM20-mCherry.RNAi}attP2; +/ + | mCherry/ Control RNAi | BDSC 35785 |
| y[1] sc[*] v[1] sev[21]; P{y[+t7.7]; +/ +; v[+t1.8] =TRiP.HMC03287}attP2/ TM3, Sb[1] Ser[1]; +/ + | Sdc RNAi 1 | BDSC 51723 |
| +/ +; UAS-Syndecan RNAi (KK); +/ +; +/ + | Sdc RNAi 2 | VDRC 107320 |
| +/ +; +/ +; UAS-Syndecan RNAi (GD); +/ + | Sdc RNAi 3 (also referred to as Sdc RNAi where only one Sdc RNAi line is used) | VDRC 13322 |
| y[1] w[*]; P{w[+mC]=UAS-Sdc.J}3; +/ +; +/ + | Sdc-J | BDSC 8564 |
| y[1] w[1118]; +/+; P{w[+mC]=UAS-mCD8::GFP.L}LL6, P{w[+mC]=tubP-GAL4}LL7/TM3, Sb[1] | *tub-GAL4* | BDSC 30030 |
| w[1118]; +/+; +/+; +/+ | *w^1118*/ Control | Celia Garcia Cortes |
| w; UAS-DIAP1/ Cyo, Kr-GAL4, UAS-GFP; TM2/ TM6, Df-YFP; +/ + | DIAP1 | Jean-Paul Vincent |
| w; UAS-DIAP1/ CyO; UAS-Sdc RNAi (VDRC 13322)/ TM6c, Df-YFP; +/ + | Sdc RNAi, DIAP1 | Kolahgar lab |
| +/ +; UAS-AuroraB-RNAi (KK); +/ +; +/ + | AuroraB RNAi | VDRC 104051 |
| y[1] sc[*] v[1] sev[21]; P{y[+t7.7] v[+t1.8]=TRiP.HMS02172}attP40; +/ +; +/ + | Klaroid RNAi | BDSC 40924 |
| y[1] sc[*] v[1] sev[21]; P{y[+t7.7]; +/ +; v[+t1.8]=TRiP.HMS01612}attP2; +/ + | Klarsicht RNAi | BDSC 36721 |
| y[1] sc[*] v[1] sev[21]; +/ +; P{y[+t7.7] v[+t1.8]=TRiP.HMS00632}attP2; +/ + | Msp300 RNAi 1 | BDSC 32848 |
| P{UAS-Dcr-2.D}1, w[1118]; nosP-GAL4-NGT40 | *nos-GAL4* | BDSC 25751 |
| w; Wor-GAL4, KlaroidCB04483, UAS Cherry::Jupiter/ Cyo; UAS Dicer/ TM6B,Tb | *wor-GAL4* | [47] |
| y[1] w[*]; Mi{PT-GFSTF.0}Sdc[MI10787-GFSTF.0]/ CyO; +/ +; +/ + | Sdc::GFP | BDSC 66373 |

tubulin promoter, inhibits GAL4 preventing expression of UAS-GFP, UAS-RNAi and UAS-flp-recombinase. Upon hatching, adult flies are transferred to the permissive temperature (29°C). This inactivates GAL80^ts, allowing the intestinal progenitor cell-specific *esg-GAL4* to drive expression of UAS-transgenes. The flp-recombinase excises a STOP codon between the ubiquitous actin promotor and GAL4, resulting in permanent and heritable GAL4 expression, and thus UAS-transgene expression, regardless of cell type. Thus, all cells which arise from progenitors after the temperature shift will express GFP, providing a visual readout for new cell production.

## Gut dissection and immunostaining

Midguts were dissected in 1X PBS (Oxoid, BR0014G) and fixed for 20 minutes at room temperature in fresh 4% methanol-free paraformaldehyde (Polysciences, 18814-10) diluted in PBS, 0.025% Triton X-100 (Sigma Aldrich, X100). Samples were given three 5-minute washes in 0.25% PBST and permeabilised for 30 minutes in 1% PBST. Samples were then incubated for 30 minutes at room temperature in blocking buffer (0.1% PBST, 0.1% BSA (Sigma, A2153-10G)), followed by incubation overnight at 4°C with primary antibodies diluted in blocking buffer. Samples were next given three 20-minute washes in 0.25% PBST and incubated for 2 hours at room temperature in secondary antibodies diluted in 0.25% PBST. Finally, guts were given three 20-minute washes in 0.25% PBST. Guts were mounted in Vectashield (Vector Laboratories, H-1000) on a glass slide (VWR International, SuperFrost 1.0mm, ISC 8037/1) with coverslip (Menzel-Gläser, 22x50mm#1, 12342118). The following primary antibodies were used: chicken anti-GFP, 1/1000 (Abcam, ab13970); rat anti-α-catenin, 1/20 (DSHB, D-CAT-1-s); rabbit anti-HA, 1/500 (Cell Signaling, 3724T); mouse anti-Sdc, 1/200 (Abmart 11053-1-2/C87); rabbit anti-γH2Av, 1/700 (Rockland, 600-401-914); mouse anti-LaminDm0 (Lamin B), 1/10 (DSHB, ADL84.12). Alexa-488-, 555-, and 647-conjugated secondary goat antibodies (Molecular Probes) were used. F-actin was stained with Phalloidin (Molecular Probes, A12380 or A22287). Nuclei were stained with DAPI (Molecular Probes, D1306) or Hoechst (Molecular Probes, H1399).

## Ovary dissection and immunostaining

Ovaries were dissected and collected in 1X PBS and then fixed for 25 minutes in 4% formaldehyde in 0.3 PBSTX (0.3% Triton-X in 1X PBS) at room temperature. Samples were given three 15-minute washes in 0.3% PBSTX and then incubated for one hour in blocking buffer (0.2 ug/μl BSA in 0.3% PBSTX), followed by incubation overnight at 4°C with primary antibodies and Phalloidin diluted in blocking buffer. Samples were then given three 15-minute washes in 0.3% PBSTX and incubated for 2 hours at room temperature in secondary antibodies diluted in blocking buffer. Samples were then given three 15-minute washes in 0.3% PBSTX, including Hoechst (33342) DNA stain in the first wash. Samples were mounted in Vectashield media (Vector Laboratories, H-1000). The following primary antibodies were used: mouse anti-γH2Av, 1/200 (DSHB, UNC93-5.2.1); mouse anti-α-Spectrin, 1/200 (DSHB, 3A9).

smFISH probes against *Sdc* were designed with the Stellaris Probe Designer (Biosearch Technologies), as described in S1 Table, and labelled with ATTO 633 as described in [84]. Briefly, after fixation and washes as above, ovaries were incubated in wash buffer (2X saline sodium citrate (SSC), 10% deionised formamide in nuclease-free water) for 10 minutes at room temperature. smFISH probes, primary antibodies and Phalloidin (Alexa Fluor 488 Phalloidin, ThermoFisher Scientific) were added to hybridisation buffer (2X SSC, 10% deionised formamide, 20mM vanadyl ribonucleoside complex, 0.1 mg/ml BSA, competitor (1:50 dilution of 5 mg/ml E.coli tRNA and 5 mg/ml salmon sperm ssDNA) in nuclease-free water), and ovaries were incubated in this buffer at 37°C overnight in the dark. Samples were given three 15-minute washes and incubated with secondary antibodies in wash buffer for 2 hours at room temperature. Finally, samples were washed in wash buffer, with the addition of Hoechst in one wash step, and mounted using Vectashield media (Vector Laboratories).

## Larval brain dissection and immunostaining

Seventy-two to one hundred forty-four-hour larvae were dissected in Schneider's insect medium (Sigma-Aldrich, S0146) and brains were fixed for 20 minutes in 4%

paraformaldehyde in PEM (100mM PIPES pH 6.9, 1mM EGTA and 1mM MgSO4). After fixing, the brains were washed with PBSBT (1 × PBS (pH7,4), 0.1% Triton X-100 and 1% BSA) then blocked with PBSBT for one hour. Primary antibody dilution was prepared in PBSBT and brains were incubated for 48 hours at 4°C. Brains were given four 30-minute washes with PBSBT, then incubated with secondary antibodies diluted in PBSBT at 4°C overnight. The next day, brains were given four 20-minute washes with PBST (1x PBS, 0.1% Triton X-100) and kept in Vectashield antifade mounting medium with DAPI (Vector laboratories, H1200), at 4°C. The following primary antibodies were used: chicken anti-GFP, 1/1000 (Abcam, Ab13970); mouse anti-Prospero, 1/100 (DHSB, MR1A); rat anti-Elav, 1/100 (DHSB, 7E8A10); rat anti-Deadpan, 1/50 (Abcam, Ab195172); rabbit anti-Asense, 1/500 (gift from Yuh Nung Jan); rabbit anti-γH2Av, 1/500 (Rockland. 600-401-914).

### RNA quantification

For each biological replicate (n = 3), total RNA was extracted from five third instar larvae using RNeasy Plus mini kit (Qiagen) according to manufacturer's instructions. cDNA synthesis was performed using Superscript III reverse transcriptase (Invitrogen). For each experiment, transcript levels were measured relative to Ribosomal Protein 49 (Rp49) expression, in technical triplicates using the iTaq Universal SYBR green supermix (Biorad) and the primer pairs shown below.

| | Forward primer | Reverse primer |
|---|---|---|
| Syndecan | TTCTGGCTGCTGTCATTGGC | TCCAGCGCATAGGATCCCTC |
| Rp49 | CCCAAGGGTATCGACAACAGA | CGATGTTGGGCATCAGATACT |

### Image acquisition and processing

Confocal images of fixed samples were acquired on a Leica TCS SP8 advanced confocal microscope with laser gain and power consistent within experiments. Gut overview images were acquired on a Leica M165FC microscope with Leica DFC3000G camera or EVOS M7000 (ThermoFisher Scientific). All images were processed and analysed with Fiji [85]. Imaris 3 64 7.5.2 was used to analyse neuroblast data presented in Fig 4. Figures were compiled in Adobe Illustrator.

Unless otherwise specified, all images show the R4/5 posterior midgut region. Midgut surface views are z-projections through half the gut depth. Images of single ISCs for nuclear lamina and shape qualifications are single z sections.

### Live imaging of larval neuroblasts

Live imaging experiments were performed on intact brains. Larvae expressing nuclear membrane and spindle marker (Klaroid::GFP and UAS-Cherry::Jupiter) together with UAS-Dicer, UAS-Sdc RNAi (VDRC 13322) and *worniu-GAL4* were dissected seventy-two or ninety-six hours after egg laying, respectively, in imaging medium (Schneider's insect medium mixed with 10% fetal bovine serum (FBS) (Sigma, F7524), 2% PenStrepNeo (Sigma, P4083), 0.02 mg/mL insulin (Sigma, 11070), 20mM L-glutamine (Sigma, G8540), 0.04 mg/mL L-glutathione (Sigma, G4251) and 5 mg/mL 20-hydroxyecdysone (Sigma, H5142)) warmed up to room temperature before use. Brains were then transferred onto IbiTreat micro-slide 15 well 3D (Ibidi, 81506) and imaged with a confocal spinning disc. Neuroblasts were imaged in the brain lobes, and we did not distinguish between type I and II neuroblasts. All images presented in Fig 4 are single z sections. Anaphase onset is defined as t=0.

## Live imaging and analysis of adult intestinal stem cells

Guts expressing YFP +/− UAS Sdc RNAi in ISCs (with *esgSu(H)* system) were carefully dissected 7 to 16 days after induction of GAL4 expression at 29°C in Shield and Sang M3 medium (Sigma, S3652), keeping the crop, Malpighian tubules and ovaries attached. Guts were mounted in imaging medium (Shields and Shang M3 medium), supplemented with 2% Fetal Bovine Serum, 0.5% penicillin-streptomycin (Invitrogen, 15140–122) and methylcellulose (Sigma, M0387-100G) 2.5% wt/vol, to stabilise the gut in the chamber [86] between a concanavalin-A coated coverslip and an oxygen permeable membrane and left to settle for 5 min. All guts were live imaged for 10–15 hours with a 2-minute interval between each scan comprising 20–25 sections of 1μm, on a 40X air objective using the EVOS M7000 (ThermoFisher Scientific). Image files were processed in FiJi, using Bleach correction and Stackreg plugins. Regions of interest (25 × 25μm) were cropped, upscaled to 512 × 512 pixels and reduced to 264 images (528 min). Cell shapes at specified timepoints were drawn and superposed using Adobe Illustrator.

## Image analysis and quantifications

**Cell density and cell type proportion.** The total number of epithelial cells (using nuclei as a proxy) and the total number of each cell type (identified by their expression of fluorescent protein or nuclear size (small, medium and large cells in *esg^{ts} F/O* experiment)) were manually counted using the cell counter tool in Fiji, and a ratio was calculated. Cell density was calculated as the total number of cells per 80x200μm area. Measurements were done on z projections of half the intestinal epithelium depth, except for Figs 1E, 1F and S2C, where single z slices were analysed, on both sides of the gut tube where possible.

**Posterior midgut progenitor cell retention and new cell production.** Posterior midguts were visually inspected and manually assigned to one of five phenotypic categories.

**Intestinal stem cell size and shape.** Masks of ISCs were manually drawn around cytoplasmic YFP signal with the Fiji freehand line on z-projections encompassing half the gut depth. The following parameters were calculated with the Fiji 'analyse particles' function: size, aspect ratio (defined as the major axis/ minor axis, where a value of 1.0 indicates a shape with equal dimensions in both axes, and higher values indicate increasingly elongated shapes) and form factor (defined as $4\pi$ x area/perimeter$^2$, where a value of 1.0 indicates a perfect circle and lower values indicate increasingly convoluted shapes).

**Nuclear lamina invaginations and nuclear shape.** High magnification single z confocal images of individual ISCs were acquired through the centre of the nucleus and processed with the Cell Profiler pipeline described in [87]. Information was extracted about nuclear eccentricity (defined as c/ a, where $c^2 = a^2 - b^2$, with a defined as half the length of the equivalent eclipse and b as half the width of the equivalent eclipse. A resulting value of 0 indicates a perfect circle and higher values indicate increasingly elongated shapes) and nuclear form factor (defined as $4\pi$ x area/perimeter$^2$, where a value of 1.0 indicates a perfect circle and lower values indicate increasingly convoluted shapes). To identify Lamin B which localises to the nuclear periphery versus that which forms invaginations, the nucleus is shrunk by a defined number of pixels. High Lamin B areas located outside this shrunk nucleus are defined as corresponding to the nuclear periphery, whereas high Lamin B areas located within the shrunk nucleus, in the nuclear interior, are defined as corresponding to invaginations.

**DNA damage.** ISCs were scored DNA damage positive if they possessed a bright, large γH2Av puncta colocalising with the nucleus or γH2Av staining covering a large amount of the nuclear area. The proportion of ISCs with DNA damage was calculated for each gut as the number of ISCs with DNA damage divided by the total number of ISCs.

**Neuroblast nuclear and cell size.** Areas were obtained by manually drawing around the border of each nucleus and cell using the Fiji freehand line tool. For quantification of nuclear division asymmetry, the nuclear diameter corresponding to the daughter neuroblast nucleus and to the daughter ganglion mother cell nucleus were measured using ImageJ, after completion of cytokinesis once nuclear growth is finished.

**Brain lobe area.** Areas were obtained by manually drawing around the border of each brain lobe using the Fiji freehand line tool.

## Statistical analyses

Data were plotted with GraphPad Prism 10 for Mac OS X. In all graphs, red line represents median. For pairwise comparisons, statistical significance was calculated using a Mann–Whitney test and p values <0.05 were considered statistically significant. Categorical data in Fig 3H were compared using the Chi square test. Categorical data in S2A and S2B Fig were compared using the Fisher Exact test due to the low expected frequency in some categories. Survival was assessed using the Log-rank test. Correlation in S4 Fig was tested with a Spearman nonparametric test. Statistics compare to age-matched control, unless otherwise indicated.

## Supporting information

**S1 Fig. Syndecan is enriched in intestinal stem and progenitor cells.** (A-C) Surface views of the R4/5 posterior midgut intestinal epithelium of flies carrying cell specific markers as indicated for each panel. Images are z projections through the intestinal epithelial depth, excluding the visceral muscle. DNA stain (blue) marks nuclei, anti-Sdc (white) marks Sdc protein. (TIF)

**S2 Fig. Sdc knockdown in progenitor cells, using three independent RNAi lines, causes progressive progenitor cell loss and failure in new differentiated cell production.** (A & B) Whole midgut views (i-iv) and posterior midgut zooms (i'-iv') from flies expressing control RNAi (i) or one of three Sdc RNAi lines (ii-iv) using the *esg^ts* F/O system. Anterior left, posterior right. GFP (black) marks progenitor cells and their progeny. Dashed boxes indicate zoomed area, with the colour of the box indicating the phenotypic category to which the gut was assigned. Graphs show the distribution of posterior midgut phenotypes, with the size of the coloured bar representing the proportion of guts assigned to the phenotypic category in each genotype. n = number of guts, from three replicates. Fisher exact test, for all p < 0.0001, however in (A) Sdc RNAi 1 shows significance in the opposite phenotypic direction to Sdc RNAi 2 & 3. (A') Quantification of *sdc* mRNA levels by qPCR (5 third instar larvae per biological replicate, n = 3 biological replicates per condition). Sdc RNAi lines were expressed with the ubiquitous *tub-GAL4, UAS-CD8-GFP* driver. (C) Surface views of midguts expressing control RNAi (i-iv) or Sdc RNAi 3 (i'-iv') using the *esg^ts* F/O system. DNA stain (blue) marks nuclei; GFP (green) marks progenitor cells and their progeny. Graph shows the proportion of small GFP^+ve cells (corresponding to progenitors (ISCs, EBs and pre-EEs) and newly differentiated EEs). n = number of guts, from three replicates. (D) ISC proportion in posterior midguts from flies expressing control or one of three Sdc RNAi lines for 28 days using the *esgSu(H)* system. n = number of guts, from three replicates. (TIF)

**S3 Fig. Sdc knockdown in ISCs does not compromise fly survival under unchallenged conditions.** (A) Survival during continuous feeding (unchallenged conditions). Log-rank test. (B) Surface views of control midguts, or midguts expressing Sdc RNAi using the *esgSu(H)* system. DNA stain (blue) marks nuclei of all cells; YFP (green) marks ISCs. (TIF)

**S4 Fig. DIAP1 expression does not prevent cellular changes in Sdc-depleted ISCs.** (A) ISC cytoplasmic area measured from YFP-expressing ISCs of indicated genotypes, across three biological replicates. (ISC numbers analysed: Control: n = 455, Sdc RNAi: n = 347, DIAP1: n = 530, Sdc RNAi, DIAP1: n = 480) (B-E) Correlation between ISC area and ISC form factor measured from YFP-expressing ISCs of indicated genotypes, across three biological replicates. (ISC numbers analysed: Control: n = 455, Sdc RNAi: n = 347, DIAP1: n = 530, Sdc RNAi, DIAP1: n = 480) (B) Control; (C) Sdc RNAi; (D) DIAP1; (E) Sdc RNAi, DIAP1. (F-G) ISC aspect ratio and ISC form factor shown for the subset of cells with an area>50µm². (Subset of ISC analysed: Control: n = 12, Sdc RNAi: n = 111, DIAP1: n = 90, Sdc RNAi, DIAP1: n = 93). ISCs are larger and more convoluted upon Sdc depletion, and this is not suppressed by DIAP1 expression.
(TIF)

**S5 Fig. Sdc knockdown, but not knockdown of LINC complex components, in ISCs causes these cells to acquire nuclear lamina invaginations and DNA damage.** (A) Control ISC, and ISCs expressing various RNAis. DNA stain (white) marks nuclei; YFP (green) marks ISCs; anti-Lamin B (white) marks nuclear lamina. (B) Proportion of ISCs, per gut, with DNA damage. n = number of guts, from three replicates. >100 ISCs analysed per genotype. (C) Proportion of ISCs, per gut, with DNA damage. n = number of guts, from two replicates. >185 ISCs analysed per genotype. (D) Proportion of ISCs, per gut, with DNA damage. n = number of guts, from three replicates. >400 ISCs analysed per genotype.
(TIF)

**S6 Fig. Sdc is dispensable from female germline stem cells.** (A) Schematic of female germline development. Female germline stem cells (fGSCs) are maintained in a stem cell niche at the anterior of the ovary in a structure called the germarium. fGSCs provide an excellent model for identifying factors involved in stem cell maintenance (i) and abscission (ii), with clear phenotypic readouts [88]. (B) Germaria from control and germline-specific knockdown of Sdc. White dashed line outlines germarium, yellow dashed line outlines fGSCs, yellow asterisk indicates stem cell niche. DNA stain (blue) marks nuclei; Phalloidin (white) marks F-actin; anti-α-Spectrin (green) marks spectrosome/fusome, allowing identification of fGSCs. (C) Germaria from control and germline-specific knockdown of Sdc. White dashed line outlines germaria, yellow dashed line outlines fGSCs, yellow asterisk indicates stem cell niche. DNA stain (blue) marks nuclei; Phalloidin (white) marks F-actin; anti-γH2Av (magenta) marks DNA damage. (D) smFISH to label Sdc transcripts (magenta) in control germaria. Rest of tissue labelling as in B. Note the low *Sdc* expression in fGSCs. The bottom panels show higher *Sdc* expression at later stages, in the oocyte.
(TIF)

**S7 Fig. Sdc expression in the different cell types of the larval brain.** (A-D) Sdc::GFP-expressing larvae were dissected and immunostained for GFP and indicated markers to differentiate between type-I (Ase+ve Dpn+ve, the majority of neuroblasts) and type-II (Ase-ve Dpn+ve) neuroblasts, ganglion mother cells (Elav-ve Pros+ve) and neurons (Elav+ve Pros-ve). DAPI (cyan) marks DNA. Note the cortical Sdc distribution in neuroblasts. (E) Time lapse images of a mitotic neuroblast (NB, outlined with white dashed line) expressing Sdc::GFP (top row, and green in merge) and Cherry::Jupiter which marks the mitotic spindle (middle row, and magenta in merge). Prior to metaphase, Sdc::GFP localises to the apical cortex (filled yellow arrowheads). After anaphase onset, Sdc::GFP relocalises to the cleavage furrow (empty yellow arrowheads). Position of cleavage furrow indicated by yellow dashed line. In the final panel, NB labels the self-renewed neuroblast and GMC labels the forming daughter ganglion mother cell. Time stamp is in minutes and seconds, with anaphase defined as time 0. Images are single z sections.
(TIF)

**S8 Fig. Sdc depletion in neuroblasts does not induce DNA damage and does not affect brain size.** (A-B) Brain lobes labelled with Klaroid::GFP (nuclear envelope, green), DAPI (DNA, cyan) and anti-γH2Av (DNA damage, white/magenta). Brain lobes contain control (A) or Sdc-depleted neuroblasts (B). The larvae were dissected at indicated time points. Three panels on the left represent maximal projections of z-slices encompassing a whole brain lobe, two panels on the right show a single z-slice to better visualise the nuclear envelope. Note the nuclear envelope dispersion upon Sdc knockdown. Scale bars: 5 μm. (C) Quantification of the proportion of γH2Av-positive neuroblasts. Despite a general increase in anti- γH2Av signal as larvae age (see A, B), Sdc-depleted neuroblasts do not acquire DNA damage. (D) Representative pictures of dissected brains from 5-day to 5.5-day old larvae and associated quantitative measures across two independent experiments (n=number of brain lobes). Brain lobes are outlined with dotted lines. Scale bar: 100 μm.
(TIF)

**S9 Fig. Sdc is required for intestinal stem cell survival and maintenance of their nuclear properties.**
(TIF)

**S1 Table. Details of smFISH probes against *Sdc*.**
(XLSX)

**S1 Data. Raw data for Fig 1E, 1F, 1H, 1I, 1J, and 1K.** File contains multiple tabs, representing data plotted in each panel.
(XLSX)

**S2 Data. Raw data for Fig 2B, 2D, 2E, and 2F.** File contains multiple tabs, representing data plotted in each panel.
(XLSX)

**S3 Data. Raw data for Fig 3C, 3D, 3E, 3G, and 3H.** File contains multiple tabs, representing data plotted in each panel.
(XLSX)

**S4 Data. Raw data for Fig 4C, 4F, 4G, 4H, and 4I.** File contains multiple tabs, representing data plotted in each panel.
(XLSX)

**S5 Data. Raw data for S2A, S2A', S2B, S2C, and S2D Fig.** File contains multiple tabs, representing data plotted in each panel.
(XLSX)

**S6 Data. Raw data used to plot survival curves.**
(XLSX)

**S7 Data. Raw data for S4 Fig.** File contains multiple tabs, representing data plotted in each panel.
(XLSX)

**S8 Data. Raw data for S5B, S5C, and S5D Fig.** File contains multiple tabs, representing data plotted in each panel.
(XLSX)

**S9 Data. Raw data for S8C and S8D Fig.** File contains multiple tabs, representing data plotted in each panel.
(XLSX)

## Acknowledgments

We thank colleagues listed in the Materials & Methods section; the Bloomington *Drosophila* Stock Center; the Vienna *Drosophila* Resource Center and the Developmental Studies Hybridoma Bank for reagents, and the Cambridge Advanced Imaging Facility for use of the confocal facility. We thank the Department of Genetics, University of Cambridge, Fly Facility. We thank Celia Garcia-Cortes for help with ISC live imaging; Nick Brown and Sonali Sengupta for feedback on the manuscript; and Roland Le Borgne and Buzz Baum for access to facilities and reagents for larval brain experiments and advice to CR.

## Author contributions

**Conceptualization:** Buffy L. Eldridge-Thomas, Golnar Kolahgar.

**Formal analysis:** Buffy L. Eldridge-Thomas, Jérome G. Bohere, Chantal Roubinet, Alexandre Barthelemy, Tamsin J. Samuels.

**Funding acquisition:** Buffy L. Eldridge-Thomas, Golnar Kolahgar.

**Investigation:** Buffy L. Eldridge-Thomas, Jérome G. Bohere, Chantal Roubinet, Alexandre Barthelemy, Tamsin J. Samuels.

**Methodology:** Buffy L. Eldridge-Thomas, Jérome G. Bohere, Chantal Roubinet, Tamsin J. Samuels, Felipe Karam Teixeira.

**Project administration:** Golnar Kolahgar.

**Supervision:** Golnar Kolahgar.

**Writing – original draft:** Buffy L. Eldridge-Thomas, Golnar Kolahgar.

**Writing – review & editing:** Buffy L. Eldridge-Thomas, Jérome G. Bohere, Chantal Roubinet, Alexandre Barthelemy, Tamsin J. Samuels, Felipe Karam Teixeira, Golnar Kolahgar.

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
