## [Editor Report · Decision Letter 0]

6 May 2024

Dear Dr Kolahgar,

Thank you very much for submitting your Research Article entitled 'The transmembrane protein Syndecan regulates stem cell nuclear properties and cell maintenance.' to PLOS Genetics.

The manuscript was fully evaluated at the editorial level. Based on reviewer comments you provided from Rview Commons, it is clear the reviewers appreciated the attention to an important problem, but raised some substantial concerns about the current manuscript. Your detailed revision plan seems reasonable and complete, so we would be willing to send out for review a much-revised version that adheres to your detailed revision plan. We cannot, of course, promise publication at that time. 

If you decide to revise the manuscript for further consideration at PLOS Genetics, please aim to resubmit within the next 60 days, unless it will take extra time to address the concerns of the reviewers, in which case we would appreciate an expected resubmission date by email to plosgenetics@plos.org.

If present, accompanying reviewer attachments are included with this email; please notify the journal office if any appear to be missing. They will also be available for download from the link below. You can use this link to log into the system when you are ready to submit a revised version, having first consulted our Submission Checklist .

PLOS has incorporated Similarity Check , powered by iThenticate, into its journal-wide submission system in order to screen submitted content for originality before publication. Each PLOS journal undertakes screening on a proportion of submitted articles. You will be contacted if needed following the screening process.

We are sorry that we cannot be more positive about your manuscript at this stage. Please do not hesitate to contact us if you have any concerns or questions.

Yours sincerely,

Giovanni Bosco, Ph.D.

Section Editor

PLOS Genetics

Giovanni Bosco

Section Editor

PLOS Genetics

---

## [Decision Letter · Decision Letter 1]

12 Nov 2024

PGENETICS-D-24-00494R1The transmembrane protein Syndecan is required for stem cell survival and maintenance of their nuclear properties.PLOS Genetics Dear Dr. Kolahgar, Thank you for submitting your manuscript to PLOS Genetics. After careful consideration, we feel that it has merit but does not fully meet PLOS Genetics's publication criteria as it currently stands. The manuscript has been evaluated by two of the three reviewers that analyzed the original submission.  While one of the reviewers was fully satisfied with your revisions, the other one (Reviewer #2) still feels that some additional validations are required. Therefore, we invite you to submit a revised version of the manuscript that addresses the points raised by the reviewer. Please submit your revised manuscript within 30 days Dec 12 2024 11:59PM. If you will need more time than this to complete your revisions, please reply to this message or contact the journal office at plosgenetics@plos.org. Please include the following items when submitting your revised manuscript:* A rebuttal letter that responds to each point raised by the editor and reviewer(s). You should upload this letter as a separate file labeled 'Response to Reviewers '. This file does not need to include responses to formatting updates and technical items listed in the 'Journal Requirements' section below.* A marked-up copy of your manuscript that highlights changes made to the original version. You should upload this as a separate file labeled 'Revised Manuscript with Track Changes '.* An unmarked version of your revised paper without tracked changes. You should upload this as a separate file labeled 'Manuscript '. If you would like to make changes to your financial disclosure, competing interests statement, or data availability statement, please make these updates within the submission form at the time of resubmission. Guidelines for resubmitting your figure files are available below the reviewer comments at the end of this letter. We look forward to receiving your revised manuscript. Kind regards, Pablo WappnerSection EditorPLOS Genetics Giovanni BoscoSection EditorPLOS Genetics Aimée DudleyEditor-in-ChiefPLOS Genetics Anne GorielyEditor-in-ChiefPLOS Genetics **Journal Requirements:** **Additional Editor Comments (if provided):****Reviewers' comments:** Reviewer's Responses to Questions

**Comments to the Authors:**

Reviewer #1: The authors have addressed most of my concerns.

Reviewer #2: Summary of Revisions:

In this revised manuscript the authors have addressed most of the issues raised but a few points remain.

The paper relies on data from RNAi knockdown of Sdc with the main figures using a single RNAi line and two additional lines used for validation in the supplementary figures. The authors have now added the requested statistical analysis to FigS2A confirming the significance of the knockdown effects on cell production for all three lines. However, it should be noted that at the 14-day timepoint Line 1 appears to show a significant difference in the opposite direction to the other two (increasing production of new differentiated cells) which should be reflected in the text. At 28 days the trend is the same direction (albeit far weaker for line 1), but Lines 2 and 3 are very consistent, which allays concerns about inconsistencies between lines.

The authors have also now included qPCR validation of the knockdowns showing knockdowns of around 50, 60 and 70% for lines 1, 2 and 3 respectively. However, this is based on n=1 replicate making it difficult to draw conclusions about the relative knockdown efficiencies. This data should be at least triplicated. In addition, Sdc is enriched in ISC and EBs and the quantification is performed after 14 days of knockdown, by which point the proportions of ISCs and EBs have changed significantly between lines. It is therefore unclear to what extent this represents differences in knockdown efficiency as opposed to loss of the cell types that express Sdc. Analysis of knockdown at an earlier timepoint may help address this.

While it is not implausible that there may be a critical threshold of knockdown a range of 50-70% is given as sufficient difference to explain the large difference in strength of phenotype. This may be relevant to the point raised by reviewer 3 regarding UAS dose compensation in the rescue experiments with Diap1 (Fig 2) as it suggests a relatively small suppression of SDC RNAi expression and therefore Sdc knockdown due to an extra UAS copy driving expression of DIAP1 could be sufficient to impact the phenotype and appear as a rescue.

The inclusion of knockout data either from mutant clones or cell-type specific CRISPR would significantly increase confidence in the findings.

LINC protein knockdowns. The authors have included references to confirm the validation of these lines in other studies supporting their use here.

Figure clarity. The authors have modified Fig 1C as requested to split channels and included additional markers in S1. The additional data is very helpful in clarifying the expression patterns.

Sdc as regulator. The text has been amended to reflect that Sdc “is required” rather than acting as a regulator addressing this point. This does somewhat detract from the potential breadth of impact of the findings as a regulatory role may have broader implications than a requirement.

Expression of Sdc in neuroblasts. Authors have included the requested data.

Knockdown validation in germline. The authors have included additional data to show very low-level expression of sdc RNA in the germline stem cells. It is not totally clear from the data presented that Sdc protein is expressed in the GSCs but the conclusion that it is not required there would still stand.

Remaining Points:

1. Phenotype direction of Sdc RNAi Line 1 in S2A should be acknowledged in the text.

2. Knockdown quantification for the three Sdc RNAi lines should be replicated and may be better assessed at an earlier timepoint.

3. Knockout data would strengthen the findings of RNAi experiments.

4. Rescue experiments involving DIAP1 expression require UAS-dosage balance controls.

**Have all data underlying the figures and results presented in the manuscript been provided?**

Reviewer #1: Yes

Reviewer #2: Yes

PLOS authors have the option to publish the peer review history of their article (what does this mean? ). If published, this will include your full peer review and any attached files.

**Do you want your identity to be public for this peer review?** For information about this choice, including consent withdrawal, please see our Privacy Policy .

Reviewer #1: No

Reviewer #2: No

 **Figure resubmission:** While revising your submission, please upload your figure files to the Preflight Analysis and Conversion Engine (PACE) digital diagnostic tool, https://pacev2.apexcovantage.com/. PACE helps ensure that figures meet PLOS requirements. To use PACE, you must first register as a user. Registration is free. Then, login and navigate to the UPLOAD tab, where you will find detailed instructions on how to use the tool. If you encounter any issues or have any questions when using PACE, please email PLOS at figures@plos.org. Please note that Supporting Information files do not need this step. If there are other versions of figure files still present in your submission file inventory at resubmission, please replace them with the PACE-processed versions. **Reproducibility:** To enhance the reproducibility of your results, we recommend that authors deposit laboratory protocols in protocols.io, where a protocol can be assigned its own identifier (DOI) such that it can be cited independently in the future. Additionally, PLOS ONE offers an option to publish peer-reviewed clinical study protocols. Read more information on sharing protocols at https://plos.org/protocols?utm_medium=editorial-email&utm_source=authorletters&utm_campaign=protocols

---

## [Decision Letter · Decision Letter 2]

21 Jan 2025

Dear Dr Kolahgar,

We are pleased to inform you that your manuscript entitled "The transmembrane protein Syndecan is required for stem cell survival and maintenance of their nuclear properties." has been editorially accepted for publication in PLOS Genetics. Congratulations!

Yours sincerely,

Pablo Wappner

Section Editor

PLOS Genetics

Giovanni Bosco

Section Editor

PLOS Genetics

Aimée Dudley

Editor-in-Chief

PLOS Genetics

Anne Goriely

Editor-in-Chief

PLOS Genetics

Comments from the reviewers (if applicable):

Reviewer's Responses to Questions

**Comments to the Authors:**

Reviewer #2: I am satisfied that the additional data provided to quantify knockdowns and clarify the text has addressed my main concern.

**Have all data underlying the figures and results presented in the manuscript been provided?**

Reviewer #2: Yes

PLOS authors have the option to publish the peer review history of their article (what does this mean? ). If published, this will include your full peer review and any attached files.

**Do you want your identity to be public for this peer review?** For information about this choice, including consent withdrawal, please see our Privacy Policy .

Reviewer #2: No

**Data Deposition**

http://datadryad.org/submit?journalID=pgenetics&manu=PGENETICS-D-24-00494R2

**Press Queries**
